# A Combination of Taurine and Caffeine in Stallion Semen Extender Positively Affects the Spermatozoa Parameters

**DOI:** 10.3390/cells12020320

**Published:** 2023-01-14

**Authors:** Marko Halo, Filip Tirpák, Tomáš Slanina, Katarína Tokárová, Martin Massányi, Lucia Dianová, Eva Mlyneková, Agnieszka Greń, Marko Halo, Peter Massányi

**Affiliations:** 1Institute of Applied Biology, Slovak University of Agriculture in Nitra, Tr. A. Hlinku 2, 949 76 Nitra, Slovakia; 2AgroBioTech, Slovak University of Agriculture in Nitra, Tr. A. Hlinku 2, 949 76 Nitra, Slovakia; 3Institute of Animal Husbandry, Slovak University of Agriculture in Nitra, Tr. A. Hlinku 2, 949 76 Nitra, Slovakia; 4Institute of Biology, Pedagogical University of Krakow, Podchorazych 2, 30-084 Krakow, Poland

**Keywords:** stallion, spermatozoa, taurine, caffeine, motility, viability, oxidative stress, chilled storage

## Abstract

This study was aimed to determine the impact of different taurine and caffeine combinations on the motility, viability, and oxidative markers of chilled stallion spermatozoa. Each stallion semen sample was diluted in a ratio of 1:2, with various taurine and caffeine concentrations (2.5–7.5 mg/mL taurine + 0.625–1.25 mg/mL caffeine) dissolved in a conventional extender. The control samples (CON) were prepared by diluting ejaculate only using the conventional extender. The motility was analyzed using a CASA system at different time intervals (0, 6, 12, 24, and 30 h) and the viability was evaluated using a mitochondrial toxicity test (MTT) performed at the end of the incubation at 5 °C. The liquid part of experimental samples was separated by centrifugation after 30 h of incubation and underwent the evaluation of oxidative stress via the quantification of markers ferric reducing ability of plasma (FRAP) and total oxidant status (TOS). The samples that were treated with a combination of taurine and caffeine significantly improved the motility parameters, mainly after 12, 24, and 30 h of incubation. Samples extended with combination of taurine and caffeine neither compromise viability nor alterations of redox status. The results of this study describe the combination of taurine and caffeine as an optimal supplement for improving the quality of stallion semen during chilled storage.

## 1. Introduction

The increasing trend of the artificial insemination of mares has spread over the last 30 years. Nowadays, most of the breeding studs allow this form of biotechnology [1]. Therefore, high demands are put on the quality of stallion semen extenders, which must maintain the motility and viability of spermatozoa in the insemination doses for a long time to enable a successful transport. Semen extenders should maintain the fertility of chilled semen for 24–48 h [2,3]. After that time, there is a significant decrease in motility, which negatively affects the success of fertilization [4]. The utilization of antioxidants or further protective substances as the component of semen extenders can improve semen quality parameters and fertility rates [5].

Taurine has been used in numerous animal species such as stallion [6], bull [7], boar [8], buffalo [9] and dog [10] as a potentially suitable component of the semen extender. Taurine, 2-aminoethanesulfonic acid, is an organic compound that is widely distributed in humans and animals. Its cytoprotective properties are based on the ability of detoxification, osmoregulation, membrane stabilization, and calcium homeostasis [11]. The presence of taurine was detected in various tissues, such as the central nervous system, kidneys, liver, retina, mammary gland and Leydig cells in testes [12]. It has also been found that taurine can participate in spermatogenesis, by increasing the antioxidant capacity of germ cells. Taurine in the fallopian tubes is thought to be essential for the mammalian spermatozoa capacitation and fertilization. Moreover, taurine helps to maintain the integrity of the normal acrosome and prevents the excessive production of free radicals, thus increasing motility [13].

Caffeine (1,3,7-trimethylxanthine) as a phosphodiesterase inhibitor may increase the intracellular level of cyclic adenosine-3′,5′ monophosphate (cAMP) in spermatozoa, which may affect the oxidative metabolism of spermatozoa. Elevated cAMP levels accelerate glycolysis, which generates adenosine triphosphate (ATP), which is important for sperm motility. Therefore, the treatment of ejaculate with caffeine increases ATP levels in cells, leading to an increased spermatozoa motility [14]. Caffeine as an alkaloid and its catabolic products xanthine and theobromine show antioxidant and prooxidant properties, which are considered to increase the spermatozoa motility and fertilization ability [15]. This beneficial feature of caffeine on spermatozoa has been reported by various studies for a stallion [16], boar [17], turkey [18], bovine, and caprine [19]. The present study is aimed to determine the effect of different combinations of taurine and caffeine on stallion spermatozoa motility parameters and the viability and markers of oxidative stress (OS) during storage at 5 °C.

## 2. Materials and Methods

### 2.1. Semen Collection and Processing

Ejaculates were collected from clinically healthy breeding stallions (*n* = 10) at the ages of 3–18 years, in District of Nitra, Slovak Republic, consisting of the breeds Oldenburger and Holsteiner. Stallions used in this study were housed and fed under the same conditions and the frequency of semen collection was performed three times per week during the breeding season. The stallions were handled carefully in accordance with the ethical guidelines of the Animal Protection Regulation of the Slovak Republic RD 377/12, complying with the European Union Regulation 2010/63. Experimental protocols were approved by the committee at Slovak University of Agriculture in Nitra, Slovak Republic. Semen was obtained on a regular collection schedule using a lubricated pre-warmed artificial vagina (Colorado type, Minitube, Tiefenbach, Germany) on a dummy. Only ejaculates with the required quality (minimum 50% motility and concentration of 1 × 10^9^ sperm/mL) were used for the subsequent experiments. The ejaculates from each stallion were extended immediately after collection with various taurine (Taurine ≥ 99%, Sigma Aldrich, St. Louis, MO, USA) and caffeine (Caffeine powder, Reagent Plus^®^, Sigma-Aldrich, St. Louis, MO, USA) concentrations, which were dissolved in conventional extender (2.45 g glucose, 4.9 g non-fat dry milk, 0.025 g gentamicin in 100 mL deionized water) in ratio of 1:2 to a final concentration of approximately 50 × 10^6^ cells/mL. Following the dilution, the samples were stored at 5 °C to simulate routine transport conditions. Control samples (CON) were prepared by diluting ejaculate only with conventional extender. The scheme of preparation of samples is presented in Table 1. The samples were continuously examined by CASA (Computer Assisted Semen Analysis) until the endpoint time interval. After 30 h of incubation at 5 °C, all samples were subjected to mitochondrial toxicity test (MTT) analysis to quantify the sperm viability. Later, all experimental and control samples were centrifugated at 3000× *g* for 10 min to obtain supernatant. In order to quantify FRAP and TOS, the supernatant was kept at −80 °C until analysis [20].

### 2.2. Motility Analysis

The motility analysis was carried out using a CASA system—Sperm Vision ^TM^ program (MiniTüb, Tiefenbach, Germany), equipped with a negative phase contrast microscope Olympus BX 51 (Olympus, Tokyo, Japan) with 20× magnification at incubation times 0, 6, 12, 24, and 30 h. Each sample was placed into the Makler Counting Chamber (depth 10 µm, Sefi-Medical Instruments, Haifa, Israel) and let to warm up to 37 °C for 10 s prior to the analysis [21]. Every output of the CASA system is the result of 7 sub-measurements of 7 different fields of Makler Counting Chamber. Using the stallion specific set up, the following selected parameters were evaluated: total motility (MOT; %), progressive motility (PRO; %), velocity curved line (VCL; µm/s), amplitude of lateral head displacement (ALH; µm), and beat-cross frequency (BCF; Hz), as described previously [6].

### 2.3. Viability

Viability of stallion spermatozoa was evaluated using the mitochondrial toxicity test (MTT) after 30 h of chilled storage. This colorimetric assay measures the reduction of 3-(4,5-dimetylthiazol-2-yl)-2,5-diphenyltetrazolium bromide (MTT; Sigma-Aldrich, St. Louis, MO, USA) to purple formazan particles by mitochondrial succinate dehydrogenase of intact mitochondria of living cells. Formazan was measured spectrophotometrically at wavelength of 570 nm against 620 nm as reference by a microplate spectrophotometer ELISA reader (Multiskan FC, ThermoFisher Scientific, Vantaa, Finland). The data were expressed in percentage of control [22].

### 2.4. Ferric Reducing Ability of Plasma (FRAP)

Analysis of FRAP was performed according to the method by Benzie and Strain [23] after 30 h of chilled storage. This test determines the total antioxidant power, based on the reduction of a ferric-tripyridyl triazine complex to its ferrous colored form in the presence of antioxidants. FeSO_4_·7 H_2_O solutions were used for calibration. A working reagent was added to 96-wells micro plate and a reagent blank reading was taken at 593 nm with ELISA reader (Multiskan FC, ThermoFisher Scientific, Vantaa, Finland). Subsequently, the standards and samples were added. Second absorbance was taken after 4 min and FRAP was calculated using the standard curve and expressed in FeSO_4_·7 H_2_O [20].

### 2.5. Total Oxidant Status (TOS)

TOS analysis was realized after 30 h of chilled storage. TOS analysis is based on the oxidation of ferrous ions of o-dianisidine complexes by the oxidants present in the sample of ferric ions. The process of oxidation reaction was supported by the glycerol molecules present in the reaction solution. Then, the ferric ions formed a colored complex with xylenol orange in the acidic environment of the reaction solution. The color intensity, which can be measured spectrophotometrically, corresponds to the total amount of oxidant molecules present in the sample. The assay is calibrated using H_2_O_2_, and the results are expressed as μmol H_2_O_2_/L [24]. We prepared reaction solutions 1 and 2. The reaction solution 1 consisted of 150 μmol xylenol orange disodium salt (Sigma-Aldrich, St. Louis, MO, USA), 140 mmol sodium chloride (Sigma-Aldrich, St. Louis, MO, USA), and 1.35 mol glycerol (Centralchem, Bratislava, Slovakia) in 25 mmol H_2_SO_4_ (Sigma-Aldrich, St. Louis, MO, USA). The TOS R2 was composed of 5 mmol ferrous ammonium sulfate hexahydrate (Centralchem, Bratislava, Slovakia), and 10 mmol o-dianisidine dihydrochloride (Sigma-Aldrich, St. Louis, MO, USA) in 25 sulfuric acid (Sigma-Aldrich, St. Louis, MO, USA). Standards (H_2_O_2_) and the samples were transferred in doubles to a 96-well plate in a volume of 35 μL. After the addition of 225 μL reaction in solution 1, the reference reading at 560 nm, using a microplate ELISA reader (Multiskan FC, ThermoFisher Scientific, Vantaa, Finland), was performed. After 10 min of incubation, 11 μL of reaction solution 2 was added to each well and the absorbance was evaluated at the 560 nm after 3 min of incubation [20].

### 2.6. Statistical Analysis

For the analysis, the GraphPad 9 software (GraphPad Software Inc., San Diego, CA, USA) was used. Descriptive statistical characteristics (mean and standard deviation) together with one-way ANOVA and Dunnett’s post-test were selected for statistical evaluations. The statistical significance was estimated on three levels: *** (*p* < 0.001), ** (*p* < 0.01), and * (*p* < 0.05). Results are interpreted as means and are finally expressed with SD. All obtained results were tested for normal Gaussian distribution using a D’Agostino–Pearson normality test and Shapiro–Wilk normality test. Furthermore, the relationship between spermatozoa quality parameters and markers of OS was evaluated using Pearson correlation. Significance was determined at *p* < 0.05 (a) and at *p* < 0.01 (A). Heatmaps with clustering were performed to visualize interactions (Pearson correlations coefficients-r) between spermatozoa quality parameters and markers of OS.

## 3. Results

### 3.1. The Effect of Taurine and Caffeine on the Spermatozoa Quality

In the initial time (time 0), the motility has not differed significantly in the experimental samples compared to control. The assessment at time 6 h revealed that the sample A reached a significantly (*p* < 0.05) increased motility in comparison to the control. Following 12 h, a significant increase in motility was observed in samples A, B, D, G, and H (*p* < 0.05), and E, F (*p* < 0.01), and I (*p* < 0.001). Interestingly, a significant (*p* < 0.001) motility increase was observed in all experimental samples enriched with the combination of taurine and caffeine after 24 and 30 h of incubation at 5 °C, compared to the control samples.

On the other hand, a significant (*p* < 0.001) inhibition of progressive motility was observed in all experimental samples in comparison to the control in the initial time interval. No significant effect on the progressive motility was recorded in the samples treated with taurine and caffeine at time 6 h. A significantly increased progressive motility in samples A, E (*p* < 0.05), and G (*p* < 0.01) in comparison to the control at time 12 h was detected. Reflecting the results of motility at time 24 and 30 h, the results of progressive motility at the same time intervals showed a significantly (*p* < 0.001) increased percentage (Figure 1).

The velocity curved line showed significantly decreased values in samples B, I (*p* < 0.05), F (*p* < 0.01), and C (*p* < 0.001), in comparison to the control in the initial measurement (time 0). The following time intervals (6, 12, 24, and 30 h) showed no significant effect on velocity curved line in the experimental samples enriched with taurine and caffeine compared to the control.

The addition of taurine and caffeine to the conventional extender showed a significantly decreased beat cross frequency at time 0 (*p* < 0.05 compared to D; *p* < 0.01 for A and G; *p* < 0.001 for of B, C, E, F, H, and I) and at time 6 h (*p* < 0.001 in relationship to C–I) when compared with the control. In contrast, the significant increase in beat cross frequency was detected at time 12 h in sample A (*p* < 0.05), at time 24 h in A (*p* < 0.01) and H (*p* < 0.05), and at time 30 h (*p* < 0.001) in all experimental samples. The significantly increased values of amplitude of the lateral head displacement were detected after 6 h in samples D (*p* < 0.05) as well as E–H (*p* < 0.01) and I (*p* < 0.001) compared to control. Contrariwise, a significant amplitude of spermatozoa lateral head displacement inhibiting effect was observed in samples A, D (*p* < 0.05), B, E (*p* < 0.001) and C, F, H, and I (*p* < 0.01), in comparison to the control at time 30 h (Figure 2).

The MTT test revealed no significant effect on viability after 30 h of incubation at 5 °C in stallion spermatozoa treated with taurine and caffeine (Figure 3).

### 3.2. The Effect of Taurine and Caffeine on the Extracellular Oxidative Stress

Samples extended with a combination of taurine and caffeine showed a higher ferric reducing ability of plasma (FRAP) in comparison to the control after 30 h of incubation at 5 °C, which indicates enhanced antioxidant activity. The range of FRAP level in experimental samples fluctuated between 840.45 and 898.48 μmol Fe^2+^, where the values of control samples were lower (827.07 μmol Fe^2+^). Experimental samples showed significant results compared to the control sample.

The results of the total oxidant status (TOS) in the experimental samples revealed very balanced values compared to the control, which had no significant effect. TOS values varied in between 106.38–112.38 μmol H_2_O_2_. The results of the extended samples revealed no significant effect on extracellular oxidative stress parameters (Figure 4).

### 3.3. Correlation Analyses

The correlation analyses between oxidative markers and motility parameters are displayed in Figure 5 and Figure 6. The results of FRAP show a neutral or moderate effect on total and a progressive motility of stallion spermatozoa. As supposed, the negative correlation was observed between the TOS and motility. However, statistically significant correlations were not observed.

## 4. Discussion

Based on previous studies that monitor the effects of taurine and caffeine separately, it is hypothesized that a combination of these substances could improve the movement parameters and viability of stallion spermatozoa during a long time period, which is desirable for the maintenance of the fertilization ability during chilled storage. The present work and the used concentrations follow up on our previous studies [6,16].

During spermatozoa storage, the reactive oxygen species (ROS) formation occurs naturally, and the integrity of spermatozoa is disturbed, which is moderated by taurine and thus allows caffeine to impact the spermatozoa motility [25]. Excessive amount of ROS results in structural damages to the spermatozoa, DNA fragmentation, and reduced motility. These effects clearly affect the long-term storage of spermatozoa. Therefore, the addition of an appropriate antioxidant to insemination doses prevents excessive ROS production and alleviates their impact on spermatozoa [26].

In this study, adding a suitable combination of taurine and caffeine did not lead to a negative impact on the observed oxidative markers, and no significant improvement of spermatozoa viability was detected. Furthermore, Zhang et al. [27] found that the optimal taurine concentration added to the semen extender during the preservation of Hu sheep semen at room temperature was 20 mM. These results indicated that 20 mM of taurine could alleviate the damage caused by ROS, which agrees with our findings.

The assessment of the total antioxidant status in buffalo cryopreserved spermatozoa extended with 50 mM of taurine measured by FRAP assay had significantly higher values compared to the control [9]. The taurine concentration corresponds with the ranges of concentration used in our study.

Moreover, the addition of taurine (25, 50, and 100 mM) to mithum (*Bos frontalis*) spermatozoa improved the antioxidant enzyme content and viability, which was similar to our investigation; however, taurine was added after the thawing of semen [28].

Through the activation of cAMP production, Chavda et al. [29] confirm that caffeine can be used as a suitable component of the semen extender for buffalo. Furthermore, caffeine at the concentration 1 and 3 mM had a positive impact on viability and reduced oxidative damage.

Our results show that combinations of different concentrations of taurine and caffeine significantly improve the total and progressive motility after 12, 24, and 30 h of storage, compared to the control sample that lacks these substances. Our findings correspond to those stated by Ramirez-Perez et al. [30]. However, we used a conventional semen extender in comparison with a commercial semen extender used in study by Ramirez-Perez et al. [29].

The importance of treating the semen with antioxidants was confirmed by Alomar [31], in a study where the effects of oxidized glutathione, cysteine, and taurine were examined. It was reported that a significantly positive effect on movement parameters of goat spermatozoa was monitored in frozen-thawed semen. Frozen-thawed semen benefited from the antioxidant activity of taurine more than cooled semen.

Previously published studies assessed the impact of caffeine not just on the chilled stallion spermatozoa, but also on cryopreserved stallion spermatozoa. Rota et al. [32] reported that the post-thawing addition of caffeine in concentrations 5 and 10 mM increased spermatozoa motility. The findings by Rota et al. [32] concerning caffeine concentrations are consistent with our range of concentrations.

Slanina et al. [33] analyzed the effect of caffeine on the motility and viability of turkey spermatozoa during incubation temperatures of 5 °C and 41 °C. Their results indicated that caffeine concentrations 0.15625, 0.3125, 0.625, 1.25, 2.5, 5, and 7.5 mg/mL significantly affected movement parameters mainly during incubation at 5 °C. The results of Slanina et al. [33] are in accordance with our findings. The same caffeine concentrations (0.625, 1.25, and 2.5 mg/mL), with the addition of taurine in our study, helped to improve motility parameters during the long-term incubation.

The addition of taurine before cryopreservation has been studied on buffalo and cattle spermatozoa. Using phase contrast microscope after the thawing of straws with cryopreserved spermatozoa, Kumar et al. [34] observed that the addition of taurine (50 mM) increased the post-thaw motility. The concentration of taurine used in the study by Kumar et al. [33] correlates with the range of taurine concentrations used in our study.

Contrasting to our results, Paál et al. [35] used a taurine concentration (1.5; 7; 12; and 5 mM) that had no clear impact on boar spermatozoa during incubation at 4 °C, which implies the possible species-specific effect of taurine.

## 5. Conclusions

To conclude, our study demonstrated that different taurine and caffeine concentrations are suitable additives to conventional stallion semen extenders. These substances significantly improved motility parameters of stallion spermatozoa during incubation at 5 °C, which represents a high importance from the breeder’s perspective, due to the frequent use of transported insemination doses. The most suitable taurine and caffeine concentration was the highest used combination (7.5 mg/mL taurine + 2.5 mg/mL caffeine). Furthermore, a combination of taurine and caffeine shows neither a negative effect on oxidative markers nor viability.

## Figures and Tables

**Figure 1 cells-12-00320-f001:**
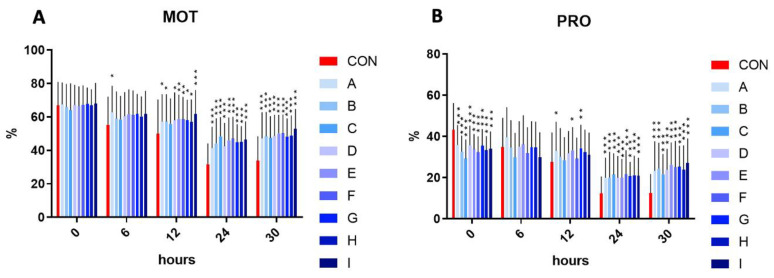
(**A**) The effect of taurine and caffeine on the total spermatozoa motility (%) at 5 °C (*n* = 10). (**B**) The effect of taurine and caffeine on the progressive spermatozoa motility (%) at 5 °C. CON: 0 mg/mL taurine + 0 mg/mL caffeine; A: 2.5 mg/mL taurine + 0.625 mg/mL caffeine; B: 2.5 mg/mL taurine + 1.25 mg/mL caffeine; C: 2.5 mg/mL taurine + 2.5 mg/mL caffeine; D: 5 mg/mL taurine + 0.625 mg/mL caffeine; E: 5 mg/mL taurine + 1.25 mg/mL caffeine; F: 5 mg/mL taurine + 2.5 mg/mL caffeine; G: 7.5 mg/mL taurine + 0.625 mg/mL caffeine; H: 7.5 mg/mL taurine + 1.25 mg/mL caffeine; I: 7.5 mg/mL taurine + 2.5 mg/mL caffeine. Each bar represents the mean value of observed samples (±SD). The level of significance was set at *** (*p* < 0.001), ** (*p* < 0.01), and * (*p* < 0.05).

**Figure 2 cells-12-00320-f002:**
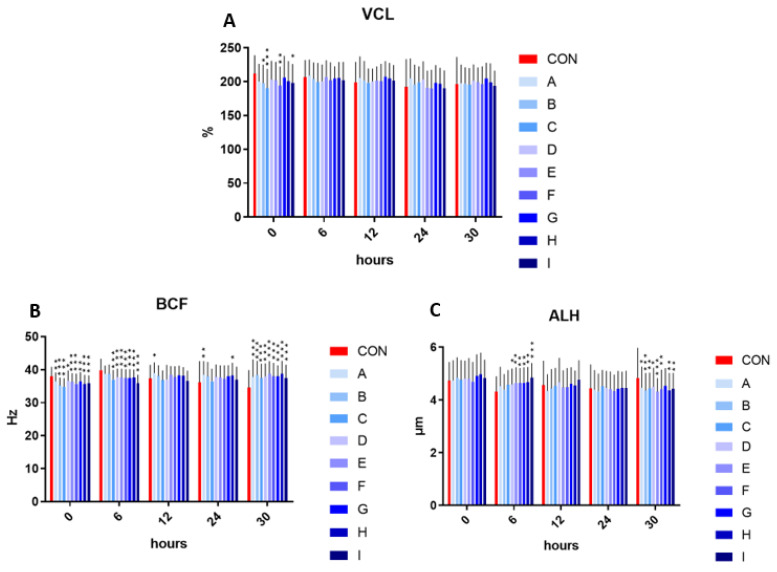
(**A**) The effect of taurine and caffeine on the velocity curved line (µm/s) at 5 °C (*n* = 10). (**B**) The effect of taurine and caffeine on the beat cross frequency (Hz) at 5 °C. (**C**) The effect of taurine and caffeine on the amplitude of spermatozoa lateral head displacement (µm) at 5 °C. CON: 0 mg/mL taurine + 0 mg/mL caffeine; A: 2.5 mg/mL taurine + 0.625 mg/mL caffeine; B: 2.5 mg/mL taurine + 1.25 mg/mL caffeine; C: 2.5 mg/mL taurine + 2.5 mg/mL caffeine; D: 5 mg/mL taurine + 0.625 mg/mL caffeine; E: 5 mg/mL taurine + 1.25 mg/mL caffeine; F: 5 mg/mL taurine + 2.5 mg/mL caffeine; G: 7.5 mg/mL taurine + 0.625 mg/mL caffeine; H: 7.5 mg/mL taurine + 1.25 mg/mL caffeine; I: 7.5 mg/mL taurine + 2.5 mg/mL caffeine. Each bar represents the mean value of observed samples (±SD). The level of significance was set at *** (*p* < 0.001), ** (*p* < 0.01) and * (*p* < 0.05).

**Figure 3 cells-12-00320-f003:**
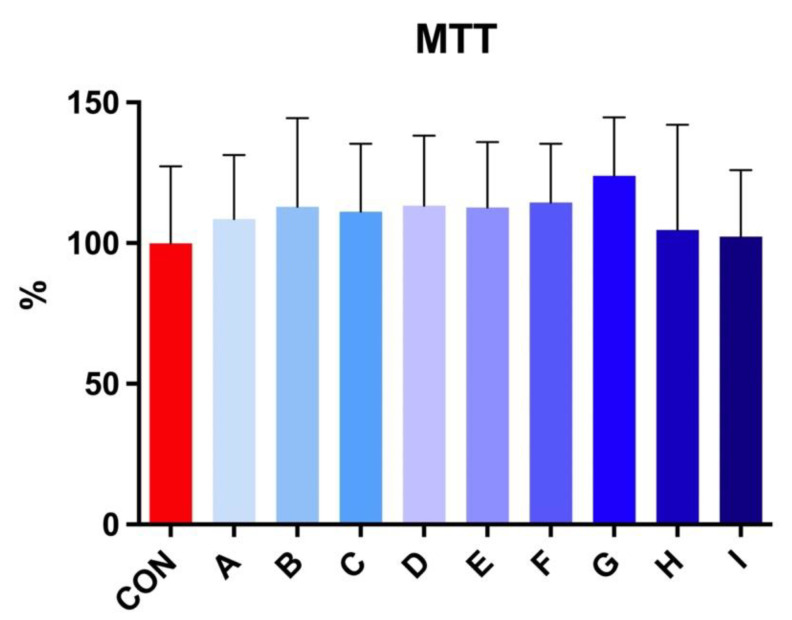
The effect of taurine and caffeine on the viability (%) of stallion spermatozoa after 30 h of incubation at 5 °C (*n* = 10). CON: 0 mg/mL taurine + 0 mg/mL caffeine; A: 2.5 mg/mL taurine + 0.625 mg/mL caffeine; B: 2.5 mg/mL taurine + 1.25 mg/mL caffeine; C: 2.5 mg/mL taurine + 2.5 mg/mL caffeine; D: 5 mg/mL taurine + 0.625 mg/mL caffeine; E: 5 mg/mL taurine + 1.25 mg/mL caffeine; F: 5 mg/mL taurine + 2.5 mg/mL caffeine; G: 7.5 mg/mL taurine + 0.625 mg/mL caffeine; H: 7.5 mg/mL taurine + 1.25 mg/mL caffeine; I: 7.5 mg/mL taurine + 2.5 mg/mL caffeine. Each bar represents the mean value of observed samples (±SD).

**Figure 4 cells-12-00320-f004:**
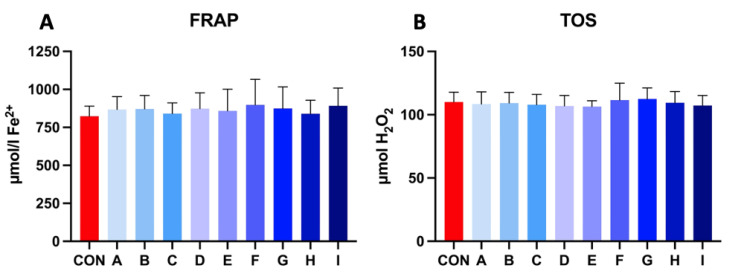
(**A**) The effect of taurine and caffeine on the ferric reducing ability of plasma (μmol Fe^2+^) of stallion spermatozoa after 30 h of incubation at 5 °C (*n* = 10). (**B**) The effect of taurine and caffeine on the total oxidant status (μmol H_2_O_2_) of stallion spermatozoa after 30 h of incubation at 5 °C. CON: 0 mg/mL taurine + 0 mg/mL caffeine; A: 2.5 mg/mL taurine + 0.625 mg/mL caffeine; B: 2.5 mg/mL taurine + 1.25 mg/mL caffeine; C: 2.5 mg/mL taurine + 2.5 mg/mL caffeine; D: 5 mg/mL taurine + 0.625 mg/mL caffeine; E: 5 mg/mL taurine + 1.25 mg/mL caffeine; F: 5 mg/mL taurine + 2.5 mg/mL caffeine; G: 7.5 mg/mL taurine + 0.625 mg/mL caffeine; H: 7.5 mg/mL taurine + 1.25 mg/mL caffeine; I: 7.5 mg/mL taurine + 2.5 mg/mL caffeine. Each bar represents the mean value of observed samples (±SD).

**Figure 5 cells-12-00320-f005:**
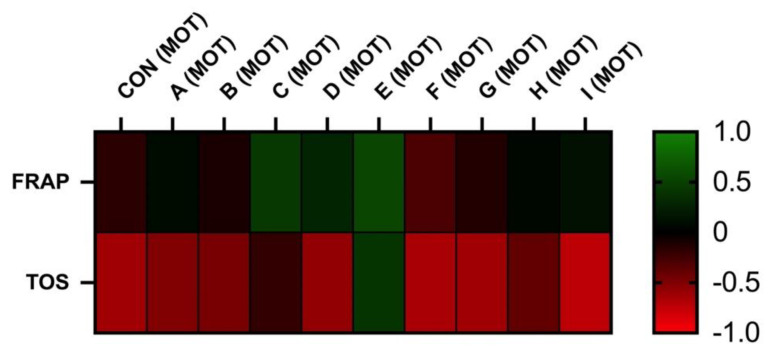
Correlations FRAP vs. MOT and TOS vs. MOT after 30 h of incubation at 5 °C (*n* = 10): correlations between FRAP and MOT were determined by comparison of respective experimental concentrations (FRAP (CON) vs. MOT (CON), FRAP (A) vs. MOT (A), FRAP (B) vs. MOT (B), FRAP (C) vs. MOT (C), FRAP (D) vs. MOT (D), FRAP (E) vs. MOT (E), FRAP (F) vs. MOT (F), FRAP (G) vs. MOT (G), FRAP (H) vs. MOT (H), and FRAP (I) vs. MOT (I)), correlations between TOS and MOT were determined by comparison of respective experimental concentrations (TOS (CON) vs. MOT (CON), TOS (A) vs. MOT (A), TOS (B) vs. MOT (B), TOS (C) vs. MOT (C), TOS (D) vs. MOT (D), TOS (E) vs. MOT (E), TOS (F) vs. MOT (F), TOS (G) vs. MOT (G), TOS (H) vs. MOT (H), and TOS (I) vs. MOT (I)).

**Figure 6 cells-12-00320-f006:**
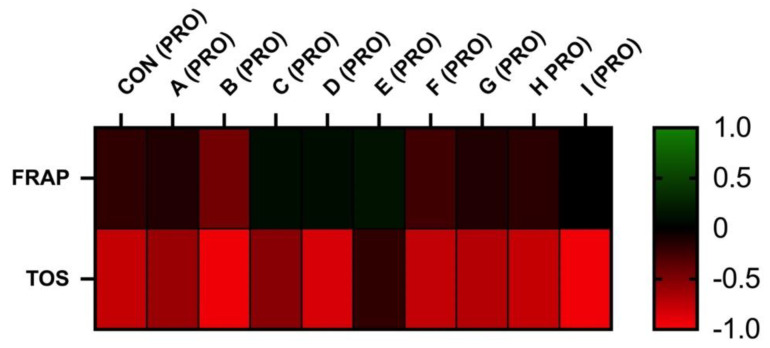
Correlations FRAP vs. PRO and TOS vs. PRO after 30 h of incubation at 5 °C (*n* = 10): correlations between FRAP and MOT were determined by comparison of respective experimental concentrations (FRAP (CON) vs. PRO (CON), FRAP (A) vs. PRO (A), FRAP (B) vs. PRO (B), FRAP (C) vs. PRO (C), FRAP (D) vs. PRO (D), FRAP (E) vs. PRO (E), FRAP (F) vs. PRO (F), FRAP (G) vs. PRO (G), FRAP (H) vs. PRO (H), and FRAP (I) vs. PRO (I)), correlations between TOS and PRO were determined by comparison of respective experimental concentrations (TOS (CON) vs. PRO (CON), TOS (A) vs. PRO (A), TOS (B) vs. PRO (B), TOS (C) vs. PRO (C), TOS (D) vs. PRO (D), TOS (E) vs. PRO (E), TOS (F) vs. PRO (F), TOS (G) vs. PRO (G), TOS (H) vs. PRO (H), and TOS (I) vs. PRO (I)).

**Table 1 cells-12-00320-t001:** Concentrations of taurine and caffeine used in the study.

Group.	Conventional Extender (mL)	Taurine (mg)	Caffeine (mg)
CON	1	0	0
A	1	2.5	0.625
B	1	2.5	1.25
C	1	2.5	2.5
D	1	5	0.625
E	1	5	1.25
F	1	5	2.5
G	1	7.5	0.625
H	1	7.5	1.25
I	1	7.5	2.5

## Data Availability

Not applicable.

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
