# Peer review of "A Combination of Taurine and Caffeine in Stallion Semen Extender Positively Affects the Spermatozoa Parameters"

_cells, 2023, doi:10.3390/cells12020320_

Round 1

Reviewer 1 Report

The work consists of a comparison of different doses of Taurine and caffeine on sperm parameters.

1- I suggest changing the title is not corresponding to the study.

2-sperm motility analysis used how many frames ?

3-why did you choosed to study kinematic parameters for VCL, ALH and BCF only?

4- metabolic activity, mitichondrial activity or viability. Choose one term please. 

5- FRAP , mitochondrial activity TOS please put the analysis times 

6-line 217 effect of taurine and caffeine on ROS: it wasn't found any significance difference in the results put it clearly .so we can not mention that the antooxidant activity were enhanced 

7-Rewrite the reults for FRAP and TOS 

8-sperm motility should compare the motility of control and caffeine taurine addition at 30h . Be consistent if you compare the viability and ROS at only 30h so do the same for motility.

Do the analysis at 0h  for control to see how the effect of caffeine and taurrine over time has on sperm parameters. 

The results founds mentionned that no significance difference was found for control and the addition of taurine and caffeine so the title do not correspond to the finding

Discussion is poor:

-need to explain more and add more other studies to compare the results of MTT, ROS and FRAP 

-The discussion is focused on motility only 

-increase in VCL and other kinematic parameters do not mean the sperm is better or the dosis have better effect on motility!!

rewrite the conclusion please.

-

Author Response

We would like to thank the reviewer for all comments and suggestions. The MS was revised according to all comments (highlighted – blue).

The work consists of a comparison of different doses of Taurine and caffeine on sperm parameters.

Q1: I suggest changing the title is not corresponding to the study.

A1: The title of the MS was modified.

Q2: sperm motility analysis used how many frames?

A2: The number of frames/fields was added to M&M section as: Every output of the CASA system is the result of 7 sub-measurements of 7 different fields of Makler Counting Chamber.

Q3: why did you chose to study kinematic parameters for VCL, ALH and BCF only?

A3: We chosen VCL, ALH and BCF, because these parameters indicate the most important results, and this should be consistent with our previous published works.

Q4: metabolic activity, mitochondrial activity, or viability. Choose one term please. 

A4: Thank you for comment. We have revised the terms in the MS.

Q5: FRAP, mitochondrial activity TOS please put the analysis times 

A5: Thank you for this comment. We added: …after 30 h of chilled storage; also, later for TOS.

Q6: line 217: effect of taurine and caffeine on ROS: it wasn't found any significance difference in the results put it clearly. So, we cannot mention that the antioxidant activity was enhanced 

A6: This statement was corrected according to reviewers’ comment. 

Q7: Rewrite the results for FRAP and TOS 

A7: This part was revised and changed according to reviewer’s comment.

Q8: sperm motility should compare the motility of control and caffeine taurine addition at 30h A8: Thank you for this comment. We analyzed motility parameters during more time intervals (0, 6, 12, 24 and 30 hours), because it is important for breeders to knows how these substances can improve or maintain motility during long time period of storage or transport. Parameters as viability, FRAP and TOS we used as supplementary.

Q9: Be consistent if you compare the viability and ROS at only 30h so do the same for motility. Do the analysis at 0h for control to see how the effect of caffeine and taurine over time has on sperm parameters.

A9: Thank you for this comment. The final statement was revised in the conclusion section.

Q10: The results found mentioned that no significance difference was found for control and the addition of taurine and caffeine, so the title does not correspond to the finding

A10: The title of MS was corrected according to reviewer’s comment.

Q11: Discussion is poor: need to explain more and add more other studies to compare the results of MTT, ROS and FRAP; the discussion is focused on motility only; increase in VCL and other kinematic parameters do not mean the sperm is better or the doses have better effect on motility!!

A11: The discussion was revised according to reviewers’ comments and some new data/reports were added.

Q12: rewrite the conclusion please.

A12: The conclusion was rewritten and revised.

Author Response

We would like to thank the reviewer for all comments and suggestions. The MS was revised according to all comments (highlighted – green).

Q1: Result and Discussion
In Figure 1 and 2, the statistical differences are difficult to view in the figures. All the figures
quality should be improved. In general, are difficult to read.

A1: Figures were adjusted. As we used many concentrations, we try to prepare the figures as clear as possible.

Q2: Line 172: Please replace the character (-) in the figure caption for the letters A-I so that it does not have the same meaning as a minus. Example: A -25 mg/mL----→ A, 25 mg/mL

A2: Figure captions were revised according to reviewers’ comment.

Q3: Line 228: please recheck. (A) for Ferric reducing ability of plasma (B) for total oxidant status. In the discussion part, please explain the mechanism of taurine and caffeine for maintaining spermatozoa quality during cold storage.

A3: This part was revised according to both reviewers and some new information were added. 

Q4: Conclusions:
Line 315: I would suggest changing "long-term" storage to "short-term"; because this research is carried out several hours or several days. Why did the authors not present the best concentration of the combination of taurine and caffeine in the conclusion?

A4: This statement was revised, and the best combined concentration was added to the conclusions.

Q5: References: The number of references is not the same as those cited in the manuscript.

A5: The references were revised.

Round 2

Reviewer 1 Report

Thank you for answering each point mentioned.